# Response thresholds alone cannot explain empirical patterns of division of labor in social insects

Yuko Ulrich[1,2�], Mari Kawakatsu[3☉], Christopher K. Tokita[4], Jonathan Saragosti[1], Vikram Chandra[1], Corina E. Tarnita[4‡*], Daniel J. C. Kronauer[1‡*]

**1** Laboratory of Social Evolution and Behavior, The Rockefeller University, New York, New York, United States of America, **2** Institute of Integrative Biology, ETH Zurich, Zurich, Switzerland, **3** Program in Applied and Computational Mathematics, Princeton University, Princeton, New Jersey, United States of America, **4** Department of Ecology and Evolutionary Biology, Princeton University, Princeton, New Jersey, United States of America

☉ These authors contributed equally to this work.
‡ CET and DJCK also contributed equally to this work.
* ctarnita@princeton.edu (CET); dkronauer@rockefeller.edu (DJCK)

**Data Availability Statement:** All simulation code is available at https://github.com/marikawakatsu/mixing-model. Additionally, software for image analysis is available at https://doi.org/10.5281/

## Abstract

The effects of heterogeneity in group composition remain a major hurdle to our understanding of collective behavior across disciplines. In social insects, division of labor (DOL) is an emergent, colony-level trait thought to depend on colony composition. Theoretically, behavioral response threshold models have most commonly been employed to investigate the impact of heterogeneity on DOL. However, empirical studies that systematically test their predictions are lacking because they require control over colony composition and the ability to monitor individual behavior in groups, both of which are challenging. Here, we employ automated behavioral tracking in 120 colonies of the clonal raider ant with unparalleled control over genetic, morphological, and demographic composition. We find that each of these sources of variation in colony composition generates a distinct pattern of behavioral organization, ranging from the amplification to the dampening of inherent behavioral differences in heterogeneous colonies. Furthermore, larvae modulate interactions between adults, exacerbating the apparent complexity. Models based on threshold variation alone only partially recapitulate these empirical patterns. However, by incorporating the potential for variability in task efficiency among adults and task demand among larvae, we account for all the observed phenomena. Our findings highlight the significance of previously overlooked parameters pertaining to both larvae and workers, allow the formulation of theoretical predictions for increasing colony complexity, and suggest new avenues of empirical study.

## Introduction

The study of collective behavior and self-organization is an active area of research across fields, from animal movement [1] to robotics [2], from tissue engineering [3] to public health [4], and from voting [5] to conservation [6]. Despite considerable theoretical and empirical

zenodo.1211644. Behavioral tracking data (raw position data and summary statistics for each individual, colony, and subcolony), as well as R scripts for statistical analyses are available in the Dryad repository https://doi.org/10.5061/dryad. hx3ffbgdd.

**Funding:** Research reported in this publication was supported by grants from the Faculty Scholars Program of the Howard Hughes Medical Institute (www.hhmi.org/programs/biomedical-research/ faculty-scholars), the Pew Biomedical Scholars Program (www.pewtrusts.org/en/projects/pew-biomedical-scholars), and the National Institute of General Medical Sciences of the National Institutes of Health (www.nih.gov/) under Award Number R35GM127007 (D.J.C.K.); Swiss National Science Foundation (www.snf.ch/en/) Advanced Postdoc Mobility (P300P3-147900) and Ambizione (PZ00P3_168066) fellowships, and a Rockefeller University Women & Science fellowship (www. rockefeller.edu/support-our-science/women-and-science/) (Y.U.); Army Research Office (www.arl. army.mil) Grant W911NF-18-1-0325 (M.K.); National Science Foundation Graduate Research Fellowship (www.nsfgrfp.org) no. DGE1656466 (C. K.T.); and a Henry and Marie-Josée Kravis Postdoctoral Fellowship (J.S.). The content is solely the responsibility of the authors and does not necessarily represent the official views of the National Institutes of Health, the Howard Hughes Medical Institute, or the Pew Biomedical Scholars Program. The funders had no role in study design, data collection and analysis, decision to publish, or preparation of the manuscript.

**Competing interests:** The authors have declared that no competing interests exist.

**Abbreviations:** DOL, division of labor; GLM, generalized linear model; LME, linear mixed effect; r.m.s.d., root–mean–square deviation.

advances, however, our understanding remains limited by a poor grasp on the impacts of heterogeneity in group composition on collective organization. This limitation stems from the difficulty in precisely controlling the sources of heterogeneity and rigorously and comprehensively measuring their impacts experimentally. This empirical challenge, in turn, has hindered the systematic testing and refining of the conceptual and theoretical frameworks employed to investigate the mechanisms underlying the collective dynamics.

The colonies of social insects are striking examples of highly integrated, complex biological systems that can self-regulate without centralized control [7]. Consequently, social insects have emerged as powerful systems to study collective behavior and social dynamics, both experimentally and theoretically [8–12]. An emergent, colony-level trait that has long been thought to depend on colony composition (e.g., in age, genotype, or morphology) is division of labor (DOL), the nonrandom interindividual variation in task performance among members of a social group that is consistent over time [13,14]. However, few experimental studies have comprehensively measured this dependence because the inherent complexity of social insect colonies usually renders their composition intractable: A typical social insect colony consists of 1 or more queens, dozens to thousands of workers of different (and often unknown) age, genotype, and morphology, and various brood development stages. This difficulty in controlling and replicating colony composition has hampered attempts to systematically test and refine the theoretical framework for collective organization in insect societies. Consequently, we have a limited understanding of how colony composition affects individual behavior and the emergent DOL, which, in turn, limits our understanding of the evolution of collective organization [15].

While several proximate mechanisms have been proposed to explain DOL in social insects (see [14] for a review), the "vast majority of studies on the impact of variability on colony behaviour have so far focused on the distribution of individual response thresholds and how this distribution affects the collective response behaviour" (see [16], p. 679). In this framework, colony members are assumed to differ in their response thresholds, i.e., in their propensity to respond to task-specific stimuli indicating the group-level demand for a given task [17–23]. Individuals with lower thresholds perform the corresponding task more readily than individuals with higher thresholds. Stimulus intensity, in turn, decreases with the number, efficiency, and time investment of individuals performing the task. With this negative feedback loop, response thresholds offer a simple mechanism for both robust and flexible allocation of individuals to tasks [14]. While refinements of response threshold models have included a self-reinforcement mechanism, whereby thresholds are modulated through experience such that individuals become more likely to perform a task that they have already performed [14,24], DOL can emerge in the absence of threshold reinforcement so long as individuals differ in their response thresholds. Indeed, the simplest version of the model, which only assumes intrinsic (i.e., fixed) variation in individual thresholds, has been successful in recapitulating certain empirically observed patterns of DOL [11,25–31].

Empirically, worker behavior in social insect colonies often correlates with individual traits [16]. For example, within a colony, workers of different age [32–35], experience [36], genotype [37] (e.g., patrilines [38,39] or matrilines [40]), or morphology (e.g., size [38,41–44]) can vary in their propensity to engage in tasks such as foraging, nursing, or nest construction. Such behavioral variation is often attributed to the developmental or genetic modulation of response thresholds. However, empirical evidence suggests that response thresholds are only one of several axes of possible individual variation. For example, workers can also vary in the efficiency with which they perform tasks [45–47] or in the average time spent performing a given task [48]. These empirical findings suggest that previously underexplored parameters may vary depending on developmental or genetic factors and may play a role in colony organization.

This possibility has led to recent calls for a diversity of parameters to be considered when investigating the relationship between colony composition and DOL [16,49,50].

Here, we combine theoretical modeling with behavioral tracking experiments in the clonal raider ant, *Ooceraea biroi*, to both assess the explanatory power of existing behavioral response threshold models and explore other axes of individual variation. The unique biology of this species affords unparalleled control over the main axes of colony composition that are thought to affect individual- and group-level behavior in social insects: genotype, age, and morphology. Specifically, colonies of clonal raider ants are naturally queenless and exclusively composed of workers that all reproduce asexually and synchronously, so that all adults within a colony are genetically almost identical and emerge in discrete age cohorts. Furthermore, individuals show variation in ovariole number that is associated with body size and other morphological features [51], making it possible to approximately sort individuals into "regular workers" (2 to 3 ovarioles) and "intercastes" (4 to 6 ovarioles) based on their size [51]. Intercastes typically represent a small fraction (3.7% to 6.3% [52]) of individuals in unmanipulated colonies, but colonies with higher fractions of intercastes (50% or more) do occur occasionally and are functional [51]. Conveniently, workers of different clonal genotypes, age cohorts, and morphologies can be mixed to create functional chimeric experimental colonies [51]. Additionally, colony behavior is controlled by larvae [27,53,54], which solicit food and care from the workers and induce them to forage. This means that colony-level task demand can be standardized or manipulated across colonies by controlling larvae number or, potentially, genotype. Finally, while colonies collected in the field contain between approximately a dozen and several hundred workers [55–57], smaller colonies of approximately 10 workers have high fitness and show complex collective behavior (e.g., group raiding [58], stable DOL, and phasic reproduction [27]) in the laboratory. Taking advantage of these features, we quantify individual and collective behavior of *O. biroi* in response to precise, independent manipulations of colony genetic, morphological, and demographic composition, as is uniquely possible in this system.

## Results and discussion

### Theoretical model

We adopt the simplest and most commonly employed formulation of the response threshold model, which assumes that individual thresholds do not change over time [18]. We consider a colony of $n$ individuals, $N_X$ of which are of type X and $N_Y = n - N_x$ are of type Y. Types X and Y represent any pair of the experimentally manipulated subcolony compositions (i.e., genotypes A and B, Young and Old, or Regular Workers and Intercastes). The colony must perform $m$ tasks; for consistency with the experimental approach (see below), we assume that there are 2 tasks ($m = 2$). At a given time step, an individual can be either performing one of the $m$ tasks (active) or not performing any (inactive). The task state of individual $i$ at time $t$ is given by the binary variable $x_{ij,t}$: If individual $i$ is active and performing task $j$ at time t, then $x_{ij,t} = 1$ and $x_{ij',t} = 0$ for all $j' \neq j$; if individual $i$ is inactive and resting, then $x_{ij,t} = 0$ for all $j$.

Each task $j$ has an associated stimulus $s_{j,t}$, signaling the group-level demand for that task. The stimulus for a task changes depending on the rate at which the demand increases (e.g., the demand for foraging increases due to increased hunger in the colony), the efficiency with which workers perform the task (e.g., more efficient foragers decrease hunger faster), and the number of individuals performing the task. Mathematically, the stimulus $s_{j,t}$ is governed by Eq (1):

$$s_{j,t+1} = s_{j,t} + \delta_j - \frac{\alpha_j^X n_{j,t}^X + \alpha_j^Y n_{j,t}^Y}{n}, \tag{1}$$

where $\delta_j$ is the task-specific demand rate, taken to be constant over time; $\alpha_j^X$ (respectively, $\alpha_j^Y$) is the task-specific performance efficiency (i.e., the rate with which an individual decreases stimulus intensity by performing the corresponding task) of type X (respectively, type Y) individuals; and $n_{j,t}^X = \sum_{i=1}^{N_X} x_{ij,t}$ and $n_{j,t}^Y = \sum_{i=N_X+1}^{n} x_{ij,t}$ are the numbers of type X and Y individuals performing task $j$ at time $t$, respectively. We assume that individuals $i = 1, \ldots, N_X$ are of type X and individuals $i = N_X + 1, \ldots, n$ are of type Y.

Each individual $i$ is assumed to have an internal threshold for each task $j$, $\theta_{ij}$, drawn at time $t = 0$ from a normal distribution with mean $\mu_j$ and normalized standard deviation $\sigma_j$ (i.e., expressed as a fraction of the corresponding mean $\mu_j$). Thus, an individual can, and typically does, have different thresholds for different tasks. Although thresholds may change over the individuals' lifetime [59], they are assumed to be fixed over the timescale of the experiments and, consequently, over the simulation runs. We refer to $\mu_j$ as the mean task threshold (or mean threshold) and to $\sigma_j$ as the threshold variance for task $j$; each can be type and/or task specific (i.e., $\mu_j^X$, $\mu_j^Y$, $\sigma_j^X$, $\sigma_j^Y$).

At each time step, inactive individuals assess the $m$ task stimuli in a random sequence until they either begin performing a task or have encountered all stimuli without landing on a task. For each encountered stimulus, individual $i$ evaluates whether to perform the task by comparing the stimulus level to its internal threshold. Specifically, given stimulus $s_{j,t}$ and internal threshold $\theta_{ij}$, individual $i$ commits to performing task $j$ with probability

$$P_{ij} = \frac{s_{ij}^{\eta}}{s_{ij}^{\eta} + \theta_{ij}^{\eta}}, \tag{2}$$

where parameter $\eta$ governs the steepness of this response threshold function. The larger the value of $\eta$, the more deterministic the behavior; in the limit $\eta \to \infty$, the response function becomes a step function. Active individuals spontaneously quit their task with a constant quit probability $\tau$. Active individuals can neither evaluate stimuli nor switch tasks without first quitting their current task.

Each agent-based simulation began with both stimuli set to 0 (i.e., $s_{j,t} = 0$ for $j = 1, 2$) and lasted $T = 10,000$ time steps (see S1 Table for parameter settings).

## Baseline model predictions

To establish baseline predictions for ant colonies with different compositions, we use the simplest implementation of this model, which assumes that ant types differ only in mean response threshold [18]. We simulated colonies that were either homogeneous (pure), with a single type of ant, or heterogeneous (mixed), with two types in equal proportions. The individual thresholds for each type were drawn from a normal distribution with the type-specific mean ($\mu_j^X = \mu^X$ or $\mu_j^Y = \mu^Y$). All other model parameters—task performance efficiency, demand rate, and threshold variance—were constant across types. Thus, the only source of heterogeneity in pure colonies was the distribution of individual response thresholds, while in mixed colonies that heterogeneity was compounded by differences in the means of the type-specific distributions.

To quantify individual behavior, we computed each individual's task performance frequency for each task, defined as the fraction of time that an individual spent performing a given task. For example, if an ant spent 2,000 time steps performing task 1 (e.g., foraging), 4,000 performing task 2 (e.g., nursing), and the remaining 4,000 being inactive in a simulation of 10,000 time steps, then it had a task 1 performance frequency of 0.2 and a task 2 performance frequency of 0.4. To quantify the mean behavior of ants in a given colony for a given task, we then averaged the individual task performance frequencies for that task across all individuals in that colony. In a mixed colony, we also quantified the type-specific mean behavior

for a given task by taking the average across all individuals of a given type in the colony instead. To quantify DOL, we measured two colony-level properties: behavioral variation, defined as the standard deviation of task performance frequency across all individuals in a colony; and specialization, defined as the mean correlation in individual task performance frequencies across time, measured as the Spearman rank correlation on consecutive windows of 200 time steps. Thus, specialization measures how consistent ants in a colony are in their task performance relative to each other.

In pure colonies, there is a single normal distribution of individual thresholds for a given task. In contrast, mixed colonies have a bimodal distribution of thresholds for each task, with the thresholds of the two types clustered around the different modes. This wider distribution of thresholds resulted in both greater behavioral variation (because individuals from the lower end of the distribution for a task are more sensitive to the stimulus for that task, they tended to perform that task more often than those from the higher end) and greater colony-level specialization (those performing a task in a given time step are likely to be from the lower end of the distribution and therefore also likely to be performing that task in a future time step) relative to pure colonies, resulting in more pronounced DOL (Fig 1A and 1B). However, all colonies, irrespective of their composition, had the same mean behavior (Fig 1C). This is because while colonies may differ in how they allocate workers to tasks—within mixed colonies, the two ant types differed in their mean task performance because the type with the lower average threshold for a given task took up that task more often than the other type—they must perform the same amount of work overall to satisfy a given demand. Thus, on average, colony members spent the same fraction of time performing each task across pure and mixed colonies.

In summary, the simple model predicted that (P1) mixed colonies would exhibit higher overall DOL but that (P2) all colonies would have the same mean behavior (Fig 1C), although (P3) the two types would diverge behaviorally in mixed colonies (Fig 1C). The same predictions held if, instead of differences in the means of the response thresholds, we assumed differences in the variances (S1 Fig).

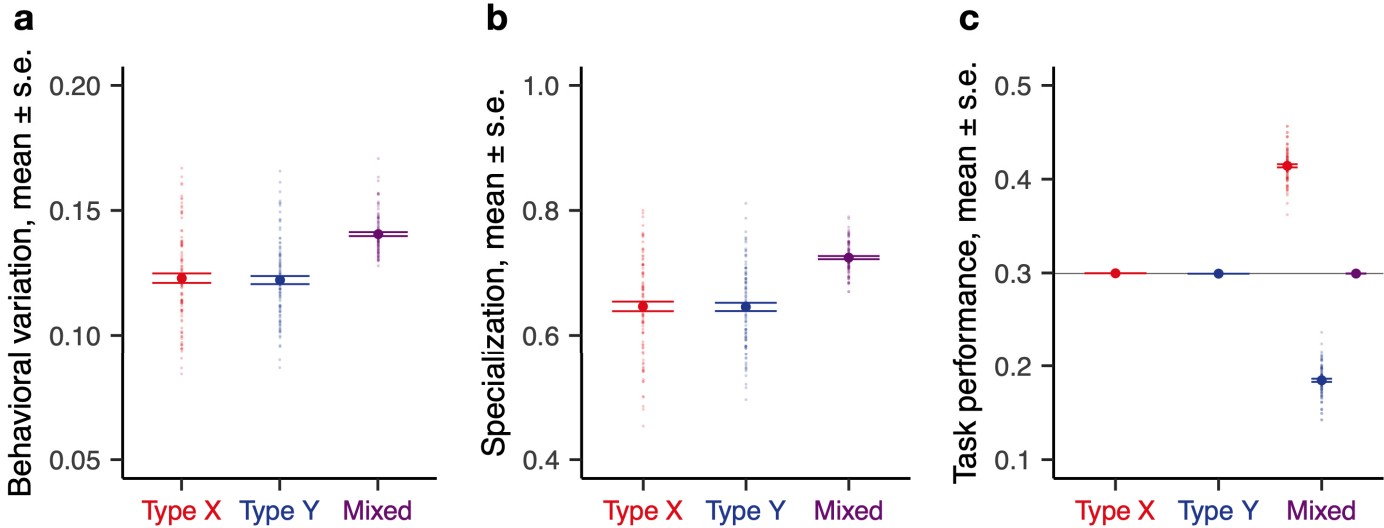

**Fig 1. Baseline theoretical predictions.** Division of labor (DOL, measured by colony-level behavioral variation (**a**), colony-level specialization (**b**)) and task performance frequency for a single task (**c**) shown as a function of colony composition. Opaque circles represent individual replicate colonies ($n = 100$ replicates for each composition); solid circles represent the average value across replicates; horizontal lines represent s.e.; and the horizontal gray line in (**c**) represents the average of the pure colonies (first 2 columns). Types X and Y differ in mean threshold: $\mu^X = 10$, $\mu^Y = 20$; all other parameters are identical across types (see S1 Table). Simulation code and data are available at https://github.com/marikawakatsu/mixing-model. DOL, division of labor.

## Empirical tests of the theoretical predictions

We then tested these theoretical predictions in experimental colonies that were either pure or 1:1 mixes of clonal raider ants that differed in one of 3 factors thought to influence DOL: genotype (A versus B [27,51]), age (around 3-month-old "old" ants versus 1-month-old "young" ants; the life span of workers in this species is around 1 year), and size (large intercastes versus smaller regular workers [51]) (S2 Table). Colonies contained larvae of the same genotype as the workers; in the case of genotype effects, the experiment was performed twice, once with larvae of each genotype (see Materials and methods). We analyzed individual behavior in 120 experimental colonies using automated tracking [27].

Because work in insect societies is spatially organized (e.g., foraging and waste disposal occur away from the nest, whereas nursing only occurs at the nest), individual spatial distribution can be used as a proxy for individual behavioral roles [60–63]. Here, the spatial distribution of each ant was measured as the two-dimensional root–mean–square deviation (r.m.s.d.) of its spatial coordinates:

$$\text{r.m.s.d.} = \sqrt{\frac{\sum_i((x_i - \bar{x})^2 + (y_i - \bar{y})^2)}{F}},$$

where $x_i$ and $y_i$ are the coordinates of the focal ant in frame $i$, $\bar{x}$ and $\bar{y}$ are the coordinates of the center of mass of the focal ant's overall spatial distribution, and $F$ is the number of frames in which the focal ant was detected. As previously shown [27], the r.m.s.d. of an ant captures its tendency to leave the nest: workers that spend a lot of time at the nest with the brood (e.g., nursing the larvae) and little time performing extranidal tasks (e.g., foraging or waste disposal) have low r.m.s.d. values, whereas workers that spend more time away from the brood have higher r.m.s.d. values (Fig 2A). Consequently, the mean r.m.s.d. of a colony reflects its collective foraging activity, as shown by the fact that r.m.s.d. increases in response to experimentally inflated nutritional demand [27]. We therefore use the r.m.s.d. as a proxy for the propensity to perform tasks away from the nest (e.g., foraging) rather than at the nest (e.g., nursing) [27]. Analogously to the simulations, we quantified the mean behavior of a given ant type as the average r.m.s.d. of all ants of that type in a colony; similarly, to quantify colony-level DOL, we computed behavioral variation as the standard deviation across r.m.s.d. values of all ants in a colony and specialization as the mean correlation in individual r.m.s.d. across time, measured as the Spearman rank correlation on consecutive days in the experiment [27] (see Materials and methods).

Colonies with different compositions often differed in mean behavior (Fig 2B–2D), inconsistent with prediction (P2). For instance, pure colonies of genotype A on average spent more time at the nest than pure colonies of genotype B (Fig 2B: $B_{pure}$ versus $A_{pure}$, LME post hoc tests: $z = 7.75$, $p = 3.64^*10^{-14}$; Fig 2C: $B_{pure}$ versus $A_{pure}$: $z = 7.45$, $p = 2.80^*10^{-13}$). Similarly, colonies of young workers spent more time at the nest than colonies of old workers (Fig 2D: $Old_{pure}$ versus $Young_{pure}$: $z = -6.05$, $p = 4.39^*10^{-09}$). That ants of different genotype [38–40] and age [33,34,64] differ in their task performance is consistent with observations in other social insects. However, such behavioral differences are theoretically only predicted to emerge (and have empirically mostly been documented) within mixed colonies (as also observed here: Fig 2B: $B_{mixed}$ versus $A_{mixed}$: $z = 4.61$, $p = 8.06^*10^{-06}$, Fig 2C: $B_{mixed}$ versus $A_{mixed}$: $z = 7.68$, $p = 6.57^*10^{-14}$, Fig 2D: $Old_{mixed}$ versus $Young_{mixed}$: $z = -13.31$, $p < 2^*10^{-16}$), and not across pure colonies (Fig 1C). Moreover, while the simple model predicted behavioral divergence between types in mixed colonies relative to pure colonies (P3), experiments produced all possible outcomes. Most surprisingly, mixing different genotypes resulted in behavioral

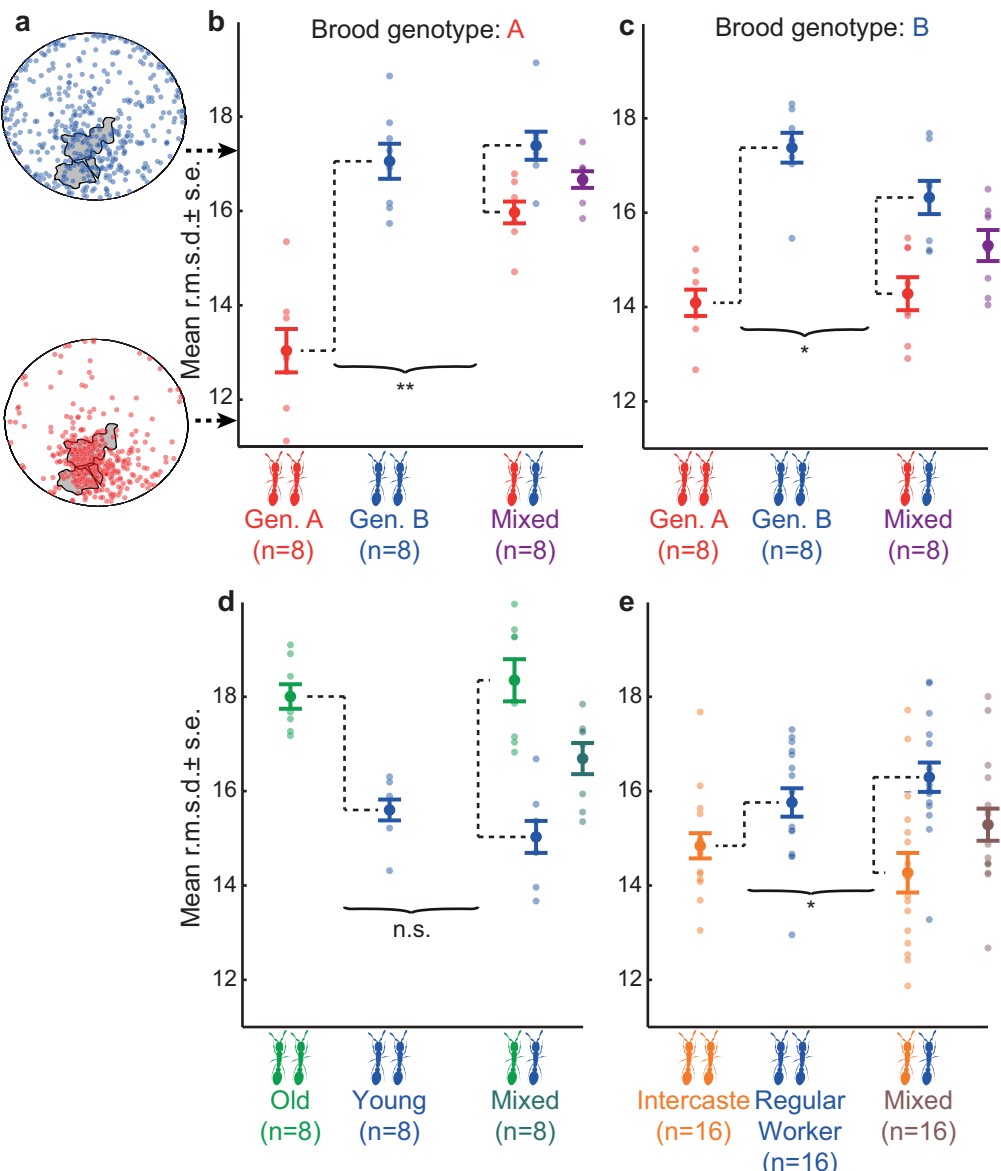

**Fig 2. Behavior as a function of colony composition.** (**a**) Spatial distributions of 2 ants with high (blue; genotype B) and low (red; genotype A) activity outside the nest. Arrows point to corresponding r.m.s.d. values. Gray areas represent the position of the larvae. (**b–e**) Mean behavior (mean r.m.s.d.) as a function of colony composition. Opaque circles represent mean behavior across individuals in replicate colonies or subcolonies. Solid circles represent average behavior across replicate colonies or subcolonies. For mixed colonies, data are shown both as type-specific and colony-level mean behavior (in "average" color). Sample sizes indicate the number of replicate colonies. Black curly brackets represent the effect of mixing on behavioral differences between types. (**b**) Behavioral convergence in genetically mixed colonies with A brood. $B_{pure} - A_{pure}$ vs. $B_{mixed} - A_{mixed}$: t test: $t = 3.86$, $p = 0.002$. (**c**) Behavioral convergence in genetically mixed colonies with B brood. $B_{pure} - A_{pure}$ vs. $B_{mixed} - A_{mixed}$: $t = 2.63$, $p = 0.025$. (**d**) No effect of mixing in demographically mixed colonies. $Old_{pure} - Young_{pure}$ vs. $Old_{mixed} - Young_{mixed}$, $t = -1.50$, $p = 0.157$. (**e**) Behavioral divergence in morphologically mixed colonies. Regular $Worker_{pure}$ − $Intercaste_{pure}$ vs. Regular $Workers_{mixed}$ − $Intercaste_{mixed}$ $t = -2.44$, $p = 0.022$. n.s., nonsignificant; *, $p < 0.05$; **, $p < 0.01$; ***, $p < 0.001$. Raw data are available at doi.org/10.5061/dryad.hx3ffbgdd. r.m.s.d., root–mean–square deviation.

convergence (see definitions in Materials and methods), whereby genotypes behaved more similarly in mixed colonies than in separation (i.e., across pure colonies) (Fig 2B and 2C). In contrast, mixing different age cohorts had no detectable effect on mean behavior (henceforth

no effect) (Fig 2D). Only mixing regular workers and intercastes produced behavioral divergence as predicted by the simple model: Intercastes spent more time at the nest than regular workers in mixed colonies (Fig 2E: Regular Worker$_{mixed}$ versus Intercaste$_{mixed}$: z = 8.95, $p < 2*10^{-16}$) but not in pure ones (Fig 2E: Regular Worker$_{pure}$ versus Intercaste$_{pure}$: z = 2.14, $p = 0.098$). Because intercastes have more ovarioles than regular workers, this behavioral difference is consistent with observations that reproductive potential often negatively correlates with the propensity to forage [52,65,66].

While in most cases mixing had symmetric effects on behavior—i.e., the behavior of both types was equally affected (Fig 2D: |Young$_{mixed}$ − Young$_{pure}$| versus |Old$_{mixed}$ − Old$_{pure}$|: $t = 0.94$, $p = 0.365$; Fig 2E: |Regular Worker$_{mixed}$ − Regular Worker$_{pure}$| versus |Intercaste$_{mixed}$ − Intercaste$_{pure}$|: $t = -0.68$, $p = 0.501$; see Materials and methods)—we found that asymmetric effects are also possible: In genetically mixed colonies with A larvae, mixing affected the behavior of A workers more than that of B workers, manifesting in asymmetric behavioral convergence (Fig 2B: |A$_{mixed}$ − A$_{pure}$| versus |B$_{mixed}$ − B$_{pure}$| $t$ test: $t = -3.86$, $p = 0.002$). Such an asymmetry was not apparent in the presence of B larvae, however (Fig 2C: |A$_{mixed}$ − A$_{pure}$| versus |B$_{mixed}$ − B$_{pure}$|: $t = 0.53$, $p = 0.607$).

Consistent with these behavioral patterns, mixed colonies had overall higher DOL than pure colonies in the age and morphology experiments, in line with the baseline model prediction (P1) (S2 and S3 Figs). However, this trend was weakened (i.e., half of the pairwise comparisons were not significant) in the genotype experiments by the emergent behavioral convergence (S2 and S3 Figs), so that mixed colonies did not systematically have higher DOL than pure colonies in all experiments, violating (P1).

Taken together, our experimental results revealed a greater diversity of behavioral patterns than predicted by the simple model: Colonies differed in mean behavior, thus violating (P2) (Fig 2); the direction and magnitude of behavioral changes in mixed colonies depended on the specific source of workforce heterogeneity, thus violating (P3) (Fig 2); and consequently, DOL was not necessarily higher in mixed than in pure colonies, thus violating (P1) (S2 and S3 Figs). Thus, heterogeneity in response thresholds alone was insufficient to explain our observations. This discrepancy prompted us to consider other biologically realistic sources of heterogeneity in the model.

## An expanded model of DOL

Previous work revealed that the developmental trajectory of *O. biroi* larvae—i.e., the size of the resulting adults—depends on nonlinear interactions between the larval genotype and the genotype of the caregiving adults [51]. This finding suggests (a) that larvae of different genotypes signal different levels of demand, e.g., for food or care; and (b) that workers of different genotypes differ in their response to a given level of larval demand, possibly via differences in their response thresholds or in the efficiency with which they perform the corresponding task. Indeed, when we added differences in task performance efficiency and in larval-induced task demand to the simple model (with between-type differences in response thresholds, i.e., consistent variation in threshold across types), we were able to qualitatively recapitulate the phenomena observed in genotype-mixing experiments (Fig 3A and 3B). Differences in task performance efficiency were, in fact, sufficient to robustly produce both colonies with different mean behaviors and behavioral convergence in mixed colonies, where the more efficient ants compensated for the less efficient ones by spending more time performing the task than they did in pure colonies. By affecting how much more the efficient ants needed to work in the mixed colonies, the differences in larval-induced task demand determined the asymmetry of the convergence. In particular, when task demand was so high that the less efficient type could

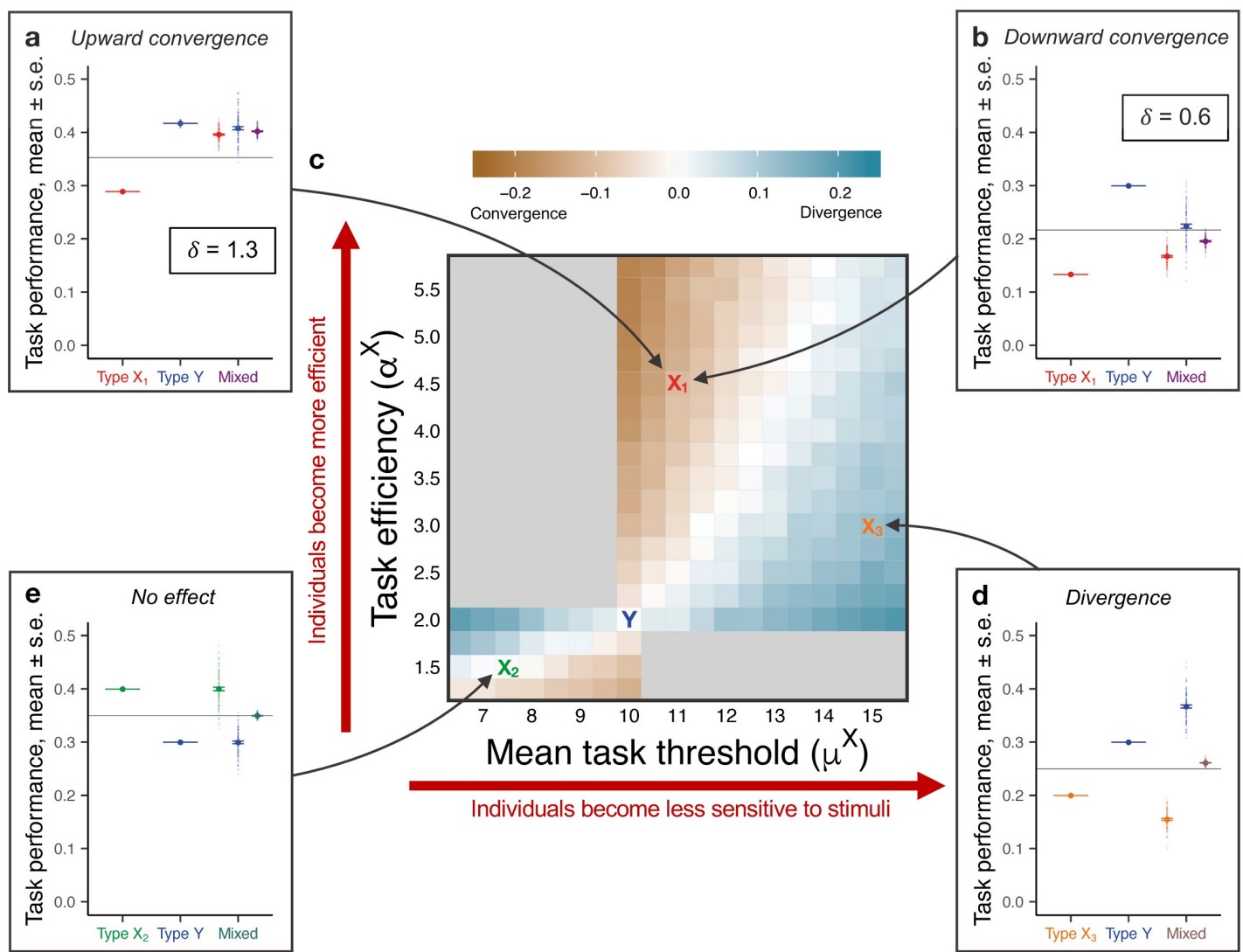

**Fig 3. Theoretical predictions of the expanded model.** (**a**, **b**, **d**, and **e**) Task performance frequency for a single task as a function of colony composition. Opaque circles represent replicate colonies ($n = 100$ replicates per composition); solid circles represent the average across replicates; horizontal bars represent s.e.; and horizontal gray lines represent the average of the pure colonies (first two columns). Identical colors indicate ants of the same type; in particular, type Y ants are the same across all panels ($\alpha^Y = 2$, $\mu^Y = 10$). (**a** and **b**) Differences in both task efficiency and mean threshold ($\alpha^{X_1} = 4.5$, $\mu^{X_1} = 11$) capture asymmetric behavioral convergence, with directionality determined by the demand rate: (**a**) upward ($\delta = 1.3$) and (**b**) downward ($\delta = 0.6$). (**d** and **e**) Differences in both task efficiency and mean threshold capture both (**d**) behavioral divergence ($\alpha^{X_3} = 3$, $\mu^{X_3} = 15$) and (**e**) a lack of effects from mixing ($\alpha^{X_2} = 1.5$, $\mu^{X_2} = 7.5$). (**c**) Change in relative task performance between mixed and pure colonies (measured as $(Y_m-X_m)-(Y_p-X_p)$) as a function of type X's efficiency and mean threshold ($n = 50$ replicates per parameter combination). Types X₁, X₂, X₃, and Y correspond to those in **a**, **b**, **d**, and **e**. Blue gradient indicates behavioral divergence ($Y_m-X_m > Y_p-X_p$); brown gradient indicates convergence ($Y_m-X_m < Y_p-X_p$); and light gray indicates regions with behavioral patterns falling outside our definitions (see Materials and methods). See S1 Table for other parameter values. Simulation code and data are available at https://github.com/marikawakatsu/mixing-model.

not keep up with the demand on their own, we recovered the experimental pattern in Fig 2B (see Fig 3A; see also stimulus dynamics in S4 Fig and S1 Text). While the simulations assumed, for simplicity, that the two tasks had the same level of demand, the analytical calculations suggest that varying demand across tasks would produce patterns qualitatively identical to either Fig 3A or Fig 3B, depending on the demand levels (see S1 Text).

Exploring the efficiency–threshold parameter space broadly recapitulated not only the behavioral convergence observed in the genotype experiments, but also the divergence and no effect patterns observed in the morphology and age experiments, respectively (Fig 3C). The

emergent pattern in mixed colonies depended on the interplay between differences in efficiency, which increased behavioral similarity, and differences in threshold, which decreased similarity. Manipulating genotypic composition corresponded to regions of the parameter space with relatively strong effects of differences in efficiency and relatively weak effects of differences in threshold (Figs 2A and 2B and 3A and 3B). Manipulating morphological composition corresponded to regions where differences in threshold had a relatively stronger effect (Figs 2E and 3D). Finally, manipulating age composition corresponded to an intermediate scenario in which the 2 effects balanced each other out (Figs 2D and 3E). Consistent with the experiments, DOL was higher in mixed colonies than in pure colonies when threshold effects were at least as strong as efficiency effects—i.e., in areas of behavioral divergence or no effect—but not when threshold effects were weaker—i.e., in areas of behavioral convergence (S5 and S6 Figs).

## Conclusions

In most social insect colonies, all factors studied here (worker genotype, age, morphology, and larval genotype) influence behavior simultaneously and in largely intractable ways. However, the unique biology of *O. biroi* allows us to break this complexity down experimentally and study each effect independently, thereby providing insight into the basic organizing principles of behavior in social groups. Our finding that the magnitude and direction of effects on DOL depend on the specific factor being manipulated underscores the importance of considering and controlling the various sources of heterogeneity that naturally act in social groups in order to study the different (and possibly opposing) effects that they have on collective organization. Moreover, our work also underscores the importance of considering factors beyond the usual suspects (e.g., age and morphology): While larval cues [67] are known to affect worker physiology [68,69] and behavior [54,70], our results highlight larvae as important players in the actual regulation of DOL between workers, something that has rarely been considered. And, on longer timescales, our findings suggest the need to consider a broader array of factors when investigating the evolution of DOL [15].

The integrated empirical and theoretical analysis reveals that models based on threshold variation alone fail to recapitulate the diverse outcomes observed in heterogeneous colonies. However, consistent with recent calls to expand theoretical investigations to other sources of heterogeneity [16,49,50], incorporating differences in larval-induced demand and in worker task performance efficiency, two parameters that, like response thresholds [13,25,71,72], are known to vary in nature [45–47,73,74], allowed us to recapitulate all empirically observed behavioral patterns. Importantly, the expanded threshold model could recapitulate these patterns using only simple individual behavioral rules and without invoking social interactions. For example, behavioral convergence—a phenomenon that intuitively appears to rely on direct social interactions—could emerge without invoking complex social processes, such as social learning [75,76] or direct information transfer between group members [77,78]. Although the theoretical treatment can only suggest candidate mechanisms, it is reassuring that the observed behaviors are robust and generic, i.e., the parameter values chosen to illustrate the versatility of the model are representative of large regions of parameter space. Nevertheless, rigorous empirical quantifications of thresholds, efficiency, and demand for realistic task–stimulus pairs—which have only rarely been attempted [48,71,79] and remain very challenging—are a critical next step toward bridging the gap between theory and empirical observations.

While we focused on the simplest model that could recapitulate our empirical results, we recognize that DOL can be influenced by an even broader set of parameters, whose roles deserve further empirical and theoretical work. For example, experience and social

interactions [28,80–82] might dynamically change individual thresholds [36] and/or task effi-ciency [34,83] over time, potentially modulating the effects observed here. It will be important to consider such effects in future theoretical extensions. At the same time, this simple model can nevertheless be used to make rich testable predictions for colonies with increasingly com-plex composition. A first attempt using different ratios of ant types led to a striking range of patterns even among the four parameter combinations in Fig 3A–3D: The model predicts that behavior can change linearly or nonlinearly as a function of colony composition depending on the between-type differences in mean threshold (S7 Fig, S1 Text). In other words, despite one type of ant being more efficient than the other in all cases considered, replacing an individual of the former with one of the latter led to proportional, greater-than-proportional, or less-than-proportional changes in task performance. Testing these predictions empirically will accelerate the productive crosstalk between theory and experiments.

Our findings add to the growing literature on the role of individual heterogeneity in the col-lective behavior of complex biological (e.g., schools of fish, neurons in a brain, pathogen strains sharing a host, etc.) and artificial (e.g., robot swarms, synthetic microbial communities, etc.) systems. Much like colonies of the clonal raider ant, these systems exhibit patterns that can be interpreted as behavioral convergence [76,78,84–86], divergence [87], and nonlinear effects of mixing on group-level phenotypes [88–90]. In turn, these patterns affect important processes such as collective decision-making [5], the transmission and evolution of disease [91,92], and the evolution of cooperative behavior [93,94]. While different variants of threshold-based models have been employed to study several of these systems [95–98], we still lack a unified theoretical framework to understand the consequences of individual differences on collective dynamics [99]. Thus, a comparative approach to the study of the basic organizing principles of heterogeneous systems across scales constitutes an important next step toward understanding the behavior of complex biological systems.

## Materials and methods

### Experimental design

Four experiments were performed to investigate the effect of genetic composition (2 experi-ments differing in the brood genotype used), age composition (1 experiment), and morpholog-ical composition (1 experiment). Each experiment comprised three treatments (2 with pure colonies, 1 with mixed colonies; S2 Table). All colonies within one experiment were monitored in parallel, but the different experiments were performed separately.

Experimental colonies were composed of workers of controlled age, genotype, and mor-phology (S2 Table), as well as larvae of controlled genotype and age. Colonies were housed in airtight Petri dishes 5 cm in diameter (corresponding to about 25 ant body lengths) with a plas-ter of Paris floor, in which the workers formed a nest by freely choosing a location where they piled their larvae. To control individual genotype, clonally related workers were sourced from the same stock colony. We used two commonly used genotypes, A and B [27,51,100,101]. To control individual age, workers were sourced from a single age cohort from the same stock col-ony. Owing to the synchronized reproduction of *O. biroi*, all age-matched workers collected this way had eclosed within a day of each other [57]. Young ants were 1 cycle old (approxi-mately 1 month old), and old ants were 3 cycles old (approximately 3 months old). The esti-mated life span of workers of this species under laboratory conditions is approximately 1 year. To control individual morphology, age-matched regular workers and intercastes from the same stock colony were screened based on body size (small or large) and the absence or pres-ence of vestigial eyes, respectively. From the time they were collected (1 to 3 days after eclo-sion) until the start of experiments, workers of a given type were kept as a group. All workers

were tagged with color marks on the thorax and gaster using oil paint markers. Experimental colonies contained 16 (genetic composition and age composition experiments) or 8 (morphological composition experiment) workers and a matching number of age-matched larvae (4 to 5 days old). This 1:1 larvae-to-workers ratio corresponds to the estimated ratio found in a typical laboratory stock colony. We used 8 (genetic composition and age composition experiments) or 16 (morphological composition experiment) replicate colonies for each group composition, for a total of 120 colonies.

Colony number and size varied across experiments due to constraints on the number of available slots in the tracking system at the time each experiment was performed. However, all colony sizes employed here were previously shown to have high fitness and stable DOL [27], and all experiments were analyzed separately so that variation in colony size should not impact the results.

The experiments took place in a climate room at 25˚C and 75% relative humidity under constant light (*O. biroi* is blind, and its behavior is not affected by light). Every three days, we cleaned and watered the plaster and added one prey item (live pupae of fire ant minor workers) per live *O. biroi* larva at a random location within the Petri dish.

## Behavioral data acquisition and analysis

Image acquisition and analysis were performed as in Ulrich and colleagues [27]. We used an automated scan sampling approach, in which a picture of each colony was acquired every approximately 400 seconds throughout the experiment by a custom setup comprising 28 webcams (B910 or C910; Logitech, Lausanne, Switzerland) and controlled LED lighting. Each webcam acquired images (5 megapixels, RGB) of 4 colonies, and the position of colonies within the setup was randomized. Custom software (available at https://doi.org/10.5281/zenodo. 1211644) was used to detect individual ants in images. For all behavioral analyses, ants were excluded from the dataset if they were detected in less than 30% of the frames acquired within the considered time frame (brood care phase or day); for ants that died during the brood care phase, the considered time frame was the portion of the brood care phase preceding death.

*O. biroi* colonies switch between reproductive phases (of approximately 18 days), in which all workers stay in the nest and lay eggs, and brood care phases (of approximately 16 days), in which workers nurse the larvae in the nest but also leave the nest to scout, forage, or dispose of waste. For each colony, behavioral analyses were restricted to the brood care phase, which started at the beginning of the experiment and ended when all larvae had either reached the nonfeeding prepupal stage or died.

For each colony or subcolony, mean behavior was computed as the average of individual r. m.s.d. values, and behavioral variation was computed as the standard deviation of individual r. m.s.d. values. Both metrics were then compared across treatments.

To quantify specialization, we use a metric appropriate for use on continuous behavioral data (here, r.m.s.d.). For each colony, specialization was defined as the Spearman correlation coefficient between individual r.m.s.d. ranks on consecutive days of the brood care phase, averaged over days. Mean rank-correlation coefficients were then compared across treatments.

## Statistical analyses

Statistical analyses were performed in R [102] separately for each of the four experiments. As the experiments were performed at different times using different cohorts of ants, we cannot rule out "batch" effects and therefore avoid any statistical analyses comparing treatments across experiments.

**Effects of individual traits on behavior.** To evaluate whether type-specific behavior depended on colony composition, we tested for a statistical interaction between the effects of individual attributes (genotypes A versus B, Young versus Old, or Regular worker versus Intercaste) and of colony composition (pure versus mixed) on individual behavior (individual r.m. s.d.) using linear mixed effects (LMEs, function *lmer* of package *lme4* [103]) models with colony as a random factor. If a significant interaction between colony composition and individual attributes was detected, we then used a second LME model with a four-level independent fixed variable combining colony composition and individual attributes ($X_p$, $Y_p$, $X_m$, and $Y_m$, where $X_k$ and $Y_k$ are the mean behavior of ant types X and Y, respectively, in pure (k = p) or mixed colonies (k = m)), followed by a Tukey post hoc test with Bonferroni–Holm correction (function *glht* of package *multcomp* [104]) for the following planned pairwise comparisons: $X_p$ versus $X_m$, $Y_p$ versus $Y_m$, $X_p$ versus $Y_p$, and $X_m$ versus $Y_m$. The two models are functionally equivalent but were used to test different hypotheses regarding interaction between terms (first model) and pairwise differences between groups (second model). When needed, the response variable was transformed (r.m.s.d.$^2$ in the genotype experiment with brood of genotype A and the age experiment, r.m.s.d$^{3/5}$ in the genotype experiment with brood of genotype B; no transformation for the morphology experiment) to satisfy model assumptions. We evaluated the significance of terms by comparing pairs of nested models using $\chi^2$ log-likelihood ratio tests following deletion of the term of interest (the interaction in the first model and the four-level variable combining colony composition and individual attributes in the second model) using the function *drop1* in R.

**Effects of genetic, demographic, and morphological mixing on DOL.** The effects of the treatment (a three-level variable: pure X, pure Y, and mixed XY) on colony-level DOL (behavioral variation and specialization) were investigated using generalized linear models (GLMs). The significance of treatment was evaluated as above. Pairwise comparisons between treatments were evaluated using Tukey post hoc tests with Bonferroni–Holm correction. Behavioral variation was square root–transformed in the genotype experiment with B larvae to satisfy model assumptions.

**Effects of genetic, demographic, and morphological mixing on behavior.** To assess how type-specific behavior was affected by mixing, and more specifically, whether the difference in behavior between types of ants was affected by mixing, we compared the difference in mean behavior (type-specific mean r.m.s.d. in each colony) between types across pure colonies to the difference in mean behavior between the same types within mixed colonies (i.e., $Y_p - X_p$ versus $Y_m - X_m$, where $Y_p > X_p$ and $Y_m > X_m$; see below for definitions of behavioral patterns), using unpaired *t* tests, after verifying assumptions of normality. In mixed colonies, the difference in mean behavior was calculated between types of ants within a colony (e.g., old and young workers from the same colony); in pure colonies, the difference in mean behavior was calculated between arbitrary pairs of pure colonies (e.g., old workers from the pure colony #1 and young ants from pure colony #1, where 1 is a replicate number assigned randomly at the beginning of the experiment). We further tested whether the amplitude of the effect differed across types by comparing the magnitude of change in type-specific behavior between pure and mixed colonies across the 2 ant types (i.e., $|X_m - X_p| \neq |Y_m - Y_p|$) with unpaired *t* tests, after verifying assumptions of normality.

## Definitions of behavioral patterns

We used the following definitions to characterize the qualitative outcomes of mixing individuals with different behavioral tendencies on individual behavior. We assume that $Y_p > X_p$ and $Y_m > X_m$, to reflect our observation that the type with higher r.m.s.d. in pure colonies always

also had higher r.m.s.d. in mixed colonies. Given this assumption, mixing could, in principle, result in one of the following patterns:

1. No effect of mixing on individual behavior: The mean behavioral difference between types across pure colonies is the same as the mean behavioral difference between types within mixed colonies, so that $Y_p - X_p = Y_m - X_m$;

2. Behavioral convergence: Individuals of different types are behaviorally more similar on average to each other when mixed, so that $Y_p - X_p > Y_m - X_m$; and

3. Behavioral divergence: Individuals of different types are behaviorally more different on average from each other when mixed, so that $Y_p - X_p < Y_m - X_m$.

## Supporting information

**S1 Text. Analytical treatment of the model.**
(PDF)

**S1 Fig. Theoretical predictions with differences in threshold variance.** Task performance frequency for a single task as a function of colony composition. Opaque circles represent individual replicate colonies ($N = 16$; $n = 100$ replicates per composition), and solid circles represent average value (± s.e.) across replicates. Horizontal gray lines represent the average of the pure colonies (first two columns). Types X and Y differ in threshold variance: $\sigma^X = 0.1$, $\sigma^Y = 0.5$; all other parameters are identical (see S1 Table). Simulation code and data are available at https://github.com/marikawakatsu/mixing-model.
(PDF)

**S2 Fig. Behavioral variation (standard deviation in r.m.s.d. across colony members) as a function of colony composition.** Small opaque circles represent individual colonies, and large solid circles represent the average values across $n$ replicate colonies. Identical colors across panels indicate ants of the same genotype, age, and morphological types. (**a**) Behavioral variation as a function of colony genetic composition in colonies with A brood ($N = 16$; $B_{pure}$ vs. Mixed: z = −2.85, $p = 0.013$; $A_{pure}$ vs. Mixed: z = 0.81, $p = 0.421$). (**b**) Behavioral variation as a function of colony genetic composition in colonies with B brood ($N = 16$; $B_{pure}$ vs. Mixed: z = −2.76, $p = 0.012$; $A_{pure}$ vs. Mixed: z = −0.81, $p = 0.419$). (**c**) Behavioral variation as a function of colony age composition ($N = 16$; $Young_{pure}$ vs. Mixed: z = 2.01, $p = 0.090$; $Old_{pure}$ vs. Mixed: z = 3.89, $p = 3.07^*10^{-04}$). (**d**) Behavioral variation as a function of colony morphological composition ($N = 8$; Regular $Worker_{pure}$ vs. Mixed: z = −2.85, $p = 0.013$, $Intercaste_{pure}$ vs. Mixed: z = 1.53, $p = 0.254$). n.s., nonsignificant; *, $p < 0.05$; **, $p < 0.01$; ***, $p < 0.001$. Raw data are available at doi.org/10.5061/dryad.hx3ffbgdd. r.m.s.d., root–mean–square deviation.
(PDF)

**S3 Fig. Colony-level specialization (day-to-day rank correlation in r.m.s.d.) as a function of colony composition.** Small opaque circles represent individual colonies, and large solid circles represent the average values across $n$ replicate colonies. Identical colors across panels indicate ants of the same genotype, age, and morphological types. (**a**) Specialization as a function of colony genetic composition in colonies with A brood ($N = 16$; GLM post hoc tests; $B_{pure}$ vs. Mixed: z = −2.78, $p = 0.017$; $A_{pure}$ vs. Mixed: z = 1.25, $p = 0.256$). (**b**) Specialization as a function of colony genetic composition in colonies with B brood ($N = 16$; $B_{pure}$ vs. Mixed: z = −2.41, $p = 0.048$; $A_{pure}$ vs. Mixed: z = 0.88, $p = 0.378$). (**c**) Specialization as a function of colony age composition ($N = 16$; $Young_{pure}$ vs. Mixed: z = 3.01, $p = 0.005$; $Old_{pure}$ vs. Mixed: z = 5.01,

$p = 1.63^* 10^{-06}$). (**d**) Specialization as a function of colony morphological composition ($N = 8$; Regular Worker$_{pure}$ vs. Mixed: $z = -4.35$, $p = 4.07^* 10^{-05}$, Intercaste$_{pure}$ vs. Mixed: $z = 2.73$, $p = 0.013$). n.s., nonsignificant; $^*$, $p < 0.05$; $^{**}$, $p < 0.01$; $^{***}$, $p < 0.001$. Raw data are available at doi.org/10.5061/dryad.hx3ffbgdd. r.m.s.d., root–mean–square deviation.
(PDF)

**S4 Fig. Dynamics of stimulus levels in pure and mixed colonies.** Each point shows the simulated stimulus level for the 2 tasks (task 1 on the horizontal axes and task 2 on the vertical axes) in the generation indicated by its color. Each of panels **a**, **b**, **d**, and **e** shows a pure colony of the type indicated; each of panels **c** and **f** shows a mixed colony of Types X and Y. Panels **a–c** ($\delta = 1.3$) correspond to Fig 3A and **d–f** ($\delta = 0.6$) to Fig 3B. (**a–c**)When the demand is higher ($\delta = 1.3$), the more efficient type (Type X) can keep up with the demand on its own (**a**) but the less efficient type (Type Y) cannot, as demonstrated by the continual growth of the stimuli (**b**); however, mixed colonies can keep up with the higher level of demand (**c**). (**d–f**) When the demand is lower ($\delta = 0.6$), the stimulus levels grow quickly at first but then stabilizes to an oscillatory pattern around a point, demonstrating that both pure and mixed colonies can keep up with the demand. Each simulation ran for 1,000 time steps; all other parameters are identical to those in the corresponding panels in Fig 3. Simulation code and data are available at https://github.com/marikawakatsu/mixing-model.
(PNG)

**S5 Fig. Theoretical predictions of the expanded model on behavioral variation.** Behavioral variation was quantified as the standard deviation of task performance frequency across individuals in a colony. Opaque circles represent individual replicate colonies ($N = 16$; $n = 100$ replicates per composition), and solid circles represent the average value (mean ± s.e.) across replicates. Types $X_1$, $X_2$, $X_3$, and Y and their corresponding parameters are identical to those in Fig 3. See S1 Table for other parameters. Simulation code and data are available at https://github.com/marikawakatsu/mixing-model.
(PDF)

**S6 Fig. Theoretical predictions of the expanded model on behavioral specialization.** Colony-level specialization was quantified using Spearman rank correlation on consecutive windows of 200 time steps. Opaque circles represent individual replicate colonies ($N = 16$; $n = 100$ replicates per composition), and solid circles represent the average value (mean ± s.e.) across replicates. Types $X_1$, $X_2$, $X_3$, and Y and their corresponding parameters are identical to those in Fig 3. See S1 Table for other parameters. Simulation code and data are available at https://github.com/marikawakatsu/mixing-model.
(PDF)

**S7 Fig. Model predictions for non-1:1 mixes.** Colonies with varying ratios of X and Y individuals were simulated under different conditions of threshold values, task performance efficiency, and task demand ($n = 100$ replicates per colony composition). Each large circle represents the mean task performance (task 1) for that mix of X and Y individuals; the neighboring smaller circles represent the means of X and Y individuals, respectively, within that mix. Dashed lines indicate the null hypothesis of linear behavioral effects of mixing types. The boxes highlight the behavioral patterns characterizing the 1:1-mixes, and their labels indicate correspondence with Fig 3 (**a**: Fig 3E; **b**: Fig 3B; **c**: Fig 3A; and **d**: Fig 3D). Types $X_1$, $X_2$, $X_3$, and Y and their corresponding parameters as in Fig 3. See S1 Table for other parameters. Simulation code and data are available at https://github.com/marikawakatsu/mixing-model.
(PDF)

**S1 Table. Parameter settings for model simulations.**
(PDF)

**S2 Table. List of experimental treatments.** Text in bold denotes the variable of interest for each experiment. All mixed colonies contained a 1:1 ratio of each ant type.
(PDF)

# Acknowledgments

We thank A. Gal for advice on analyses and O. Feinerman and M. Liu for contributions to the tracking algorithms. This is Clonal Raider Ant Project paper #13.

# Author Contributions

**Conceptualization:** Yuko Ulrich, Mari Kawakatsu, Christopher K. Tokita, Jonathan Saragosti, Corina E. Tarnita, Daniel J. C. Kronauer.

**Data curation:** Yuko Ulrich, Mari Kawakatsu.

**Formal analysis:** Yuko Ulrich, Mari Kawakatsu, Christopher K. Tokita.

**Funding acquisition:** Yuko Ulrich, Christopher K. Tokita, Jonathan Saragosti, Daniel J. C. Kronauer.

**Investigation:** Yuko Ulrich, Mari Kawakatsu, Christopher K. Tokita, Vikram Chandra, Corina E. Tarnita, Daniel J. C. Kronauer.

**Methodology:** Yuko Ulrich, Mari Kawakatsu, Christopher K. Tokita, Jonathan Saragosti.

**Project administration:** Corina E. Tarnita, Daniel J. C. Kronauer.

**Resources:** Yuko Ulrich, Mari Kawakatsu, Jonathan Saragosti, Daniel J. C. Kronauer.

**Software:** Yuko Ulrich, Mari Kawakatsu, Christopher K. Tokita, Jonathan Saragosti.

**Supervision:** Corina E. Tarnita, Daniel J. C. Kronauer.

**Validation:** Corina E. Tarnita, Daniel J. C. Kronauer.

**Visualization:** Yuko Ulrich, Mari Kawakatsu, Daniel J. C. Kronauer.

**Writing – original draft:** Yuko Ulrich, Mari Kawakatsu, Corina E. Tarnita, Daniel J. C. Kronauer.

**Writing – review & editing:** Yuko Ulrich, Mari Kawakatsu, Christopher K. Tokita, Jonathan Saragosti, Vikram Chandra, Corina E. Tarnita, Daniel J. C. Kronauer.

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
