## [Editor Report · Decision Letter 0]

15 Apr 2020

Dear Dr Kronauer, 

Thank you very much for submitting your manuscript entitled "Emergent behavioral organization in heterogeneous groups of a social insect" for consideration as a Research Article by PLOS Biology. 

We were interested to read of your work investigating the role of individual heterogeneity in determining the collective behavior of groups. Although we appreciate that the findings may offer insights useful for understanding the emergent properties of social groups, regretfully we did not find that the manuscript represented the strength of advance that we seek for publications in PLOS Biology.

While we cannot consider your manuscript further for publication in PLOS Biology, we suggest, as an alternative, that you consider transferring this manuscript to PLOS ONE (http://journals.plos.org/plosone/). 

PLOS ONE is a peer-reviewed journal that accepts scientifically sound primary research. The review process at PLOS ONE differs from other PLOS journals in that it does not judge the perceived impact of the work or whether this falls within a particular area of research. Rather, it focuses on whether the study has been performed and reported to high scientific and ethical standards, and whether the data support the conclusions. This approach helps to eliminate the rejection cycles that authors commonly encounter when submitting to one journal after another. Please note that the journals are editorially independent and we therefore cannot guarantee the outcome if you choose to pursue a transfer.

If you would like to submit your work to PLOS ONE, please click the following link:

<DeepLinkData><DeepLinkTypeID>27</DeepLinkTypeID><peopleID>252373</peopleID><userSecurityID>0791f9dd-7fed-440d-9a4c-a5083ad0bb59</userSecurityID><documentID>41088</documentID><revision>0</revision><manuscriptNumber>PBIOLOGY-D-20-00979</manuscriptNumber><docSecurityID>f418d1f2-e910-484a-b701-06634d5fbaae</docSecurityID></DeepLinkData>

If you do NOT wish to submit your work to PLOS ONE, please click this link to decline: 

<DeepLinkData><DeepLinkTypeID>28</DeepLinkTypeID><peopleID>252373</peopleID><userSecurityID>0791f9dd-7fed-440d-9a4c-a5083ad0bb59</userSecurityID><documentID>41088</documentID><revision>0</revision><manuscriptNumber>PBIOLOGY-D-20-00979</manuscriptNumber><docSecurityID>f418d1f2-e910-484a-b701-06634d5fbaae</docSecurityID></DeepLinkData>

Should you choose to transfer your submission to PLOS ONE, you will receive a confirmation email within 24-48 hours of accepting the transfer. Please note, all PLOS journals are editorially independent and vary in submission requirements. Your submission details and manuscript files will be transferred automatically; once in the PLOS ONE Editorial Manager site, your submission will be returned to you and you will be asked to provide additional information before you can finalize your new submission to PLOS ONE. If you have any questions, please feel free to contact the journal at plosone@plos.org.

Thank you for giving us the opportunity to consider your work.

Sincerely,

Roli Roberts

Senior Editor

PLOS Biology

---

## [Decision Letter · Decision Letter 1]

12 Jan 2021

Dear Daniel,

Thank you very much for submitting your manuscript "Emergent patterns and underlying mechanisms of behavioral organization in heterogeneous social groups" for consideration as a Research Article at PLOS Biology. Your manuscript has been evaluated by the PLOS Biology editors, an Academic Editor with relevant expertise, and by four independent reviewers.

You'll see that three of the reviewers were very positive about your study. Reviewer #3 feels that in its current form it may be somewhat too specialised for our journal, but we think that by addressing this specific comments made by all four of the reviewers you should be able to make the article more accessible and of broader interest. In addition, the Academic Editor has kindly provided some additional comments that you should also attend to (see the foot of this email).

In light of the reviews (below), we are pleased to offer you the opportunity to address the comments from the reviewers in a revised version that we anticipate should not take you very long. We will then assess your revised manuscript and your response to the reviewers' comments and we may consult the reviewers again.

We expect to receive your revised manuscript within 2 months.

**IMPORTANT - SUBMITTING YOUR REVISION**

Your revisions should address the specific points made by each reviewer and by the Academic Editor. Please submit the following files along with your revised manuscript:

*Resubmission Checklist*

*Published Peer Review*

*PLOS Data Policy*

*Blot and Gel Data Policy*

Sincerely,

Roli

Senior Editor,

rroberts@plos.org,

PLOS Biology

REVIEWERS' COMMENTS:

Reviewer #1:

[identifies himself as Peter Nonacs]

I very much like this paper for matching behavioral outcomes to mathematical outcomes from versions of varying complexity. Thus, when observed behavior does not match predicted behavior, the models can have biologically reasonable features added them to see what might account for the deviation. This is a very powerful integration of test and theory.

Most of my concerns here are with how meaningful are the observed behaviors under the very simplified world of the lab experiments. A lot of this might be answered by providing a fuller account of the natural history of O. biroi (as I and probably most of the readers will not be familiar with this species). For example, it is described as a "raider" ant which brings to mind colonies of thousands to millions of ants. Is this true? The authors draw very small subsets of individuals from large stock colonies. How large were these stocks? Questions:

The size of the experimental colonies ranged between 8-16 workers. How does this compare to sizes in the field? How does this species start new colonies? In short, would we expect to ever see sizes this small? If not, why would we expect "typical" behavior in the lab? There is certainly a fair amount of evidence in other species that DOL is greatly affected by worker numbers.

What were the colonies fed? I'm assuming that foraging is not raiding other species in the lab. If foraging is just going a short distance to pick up food just laying there, how might this affect DOL?

The worker vs intercaste comparison is described as just varying size. However, I assume an 'intercaste' is cross between a worker morphology and what was once in the past, a queen morphology. Given this, wouldn't one expect an intercaste to be simply less efficient at both brood and foraging? If so, their existence is better explained as group-level need to be more reproductively fecund rather than better at any task other than laying eggs. What is 'usual' frequency of intercastes in natural nests? Do you ever see any pure intercaste colonies of O. biroi? In short, I not sure how meaningful this manipulation is for explaining DOL.

How is age handled in the experiment and in the model. One of the main explanations for DOL is an age-based change in individual thresholds that goes from a bias for brood care towards doing the more dangerous foraging. In the model, it is assumed that individual thresholds are fixed over a time frame such as that of the experiment. No argument with that. However, how does model handle situations in the mixed condition? Are different ages assigned different thresholds? (In which case, it would be no surprise to produce DOL!) Overall, the experiments do seem to find patterns that age based transitions are important in creating DOL, but the authors do not seem to want to explicitly conclude that. I do realize that there are reviewers that bridle at any suggestion that age determines task, but….

Since the authors appear to know what each individual does, have they considered just analyzing the greatest outliers in each colony? Would the patterns of the 'most specialized' be more in line with predicted effects of the variables they test?

Finally, a small point. If we read the manuscript linearly, we encounter "r.m.s.d" before it is defined (in the Methods). It would be helpful to define it where it first appears in the text.

Respectfully submitted,

Peter Nonacs

Reviewer #2:

In this manuscript, the authors test the ability of an initially simple response threshold model to predict observed patterns of division of labour within colonies of a clonal queenless ant species. Division of labour is quantified using movement patterns, following the basic assumption that foragers travel far and wide while those engaged in nest-based tasks do not. The behaviour of colonies containing two "types" of ants is used to test the model, where colonies are composed entirely of one type, entirely of the second type, or a mixture of the two types. The nature of the "type" is varied- it can be two morphological variants, two age-cohort variants or two genetic variants. On finding that the simple model, which relies only on variation in response thresholds between types, does not provide a good fit to their findings, the authors expand the model post-hoc. Once variation between types in larval demand behaviour and in worker efficiency at particular tasks are incorporated, the fit is much better. Thus, we can conclude that types must vary not only in response threshold, but also in the latter two factors, to produce observed patterns of division of labour.

The system is indeed extraordinarily well-suited to the target question, as the authors claim, and the experiments appear well designed. My review focuses on the design and interpretation of the experiments that test the model predictions, rather than the development of the model itself (as an empiricist the latter is beyond my expertise). My criticisms are not major, but they mostly relate to the clarity of the manuscript, which in some places render the validity of the conclusions difficult to judge. Revision to improve the clarity in several places should make this task easier. Line by line comments are below. 

Title: Doesn't really tell me anything about the findings! This title is very abstract.

Lines 82-85: On first reading, it wasn't clear to me why the fact that there are several axes of individual variation was a problem, given that such variation could well correlate with response thresholds. After looking into the cited references, it became clearer but I'd suggest taking more time to explain your reasoning here, given that the target journal is non-specialist. 

Line 112: It would be helpful here to briefly outline a little more of the model here, in order that these predictions can be understood. While the description in the methods is clear, it isn't possible to understand this section fully without first reading that. For example, you indicate what is meant by "type" later, but it needs some explanation when first encountered. Plus, could you explain why the model predicts that mixed colonies will have more specialists? As I understand it, a specialist is an individual with a low response threshold for a particular task. In cases where the mean response threshold is higher for Type Y than Type X, across both tasks, why will there be more specialists in mixed colonies? More clarity is needed here, again, with a general audience in mind.

Line 124/293-297: Why the change in colony size from 16 to 8, and corresponding increased number of replicates, for the morphological mix? 

Line 354: Please state which response variables were transformed and which transformations were used (because it aids replication attempts based on the raw data)

Line 340- end of statistical analysis section: Did you perform model selection, and if so, how? 

Line 349: Could you explain the logic behind including both LME models, rather than simply one? I can see why you would perform the second one, if you specifically wished to evaluate the pairwise differences between groups, and perhaps you also specifically wanted a parameter estimate for the overall effect of either Pure/Mixed or Type. But if this is the case, please make your logic clear. 

Line 362-369: I struggled to understand both the aim and method behind this analysis. The stated aim is "to assess whether type-specific behaviour was affected by colony composition"- but is this not achieved by your LME model described above at line 349, and by the initial LME model too? A significant interaction effect in the initial LME would confirm that the effect of type on behaviour depended upon colony composition. And when you say "Yp-Xp vs Ym-Xm", how did you match up the colonies? For each experiment, you had 8 colonies per replicate, so presumably there are 16 data points being compared in the t-tests, e.g. Yp1- Xp1, Yp2- Xp2…. Yp8- Xp8. It's not clear how the colonies were allocated into these pairs. Apologies if I have misunderstood, but this comparison is really not at all clear to me.

Figure 2: It isn't immediately clear how to read the curly brackets because they don't always line up correctly with the data points (e.g. Figure 2b), nor are the x-axis always well aligned with the data points (could you perhaps group the "mixed" results further away from the "pure" ones? This would help in that respect). Generally, I don't find it helpful to have all the statistical results placed in the figure legend. It is not always clear which comparisons you are referring to when you make statements in the text (e.g. line 140-142). Could you put the relevant result in the text? The link between Figure 2, the reported results in the text, and the statistical comparisons, is not at all clear.

Reviewer #3:

In this work, the authors combined experimental and theoretical approaches to investigate how the heterogeneity in group composition shapes division of labor. Based on a qualitative agreement between experimental data and theoretical predictions, the authors concluded that the variation in response thresholds is not sufficient to account for the observed patterns of division of labor. 

First of all, I regret that the authors did not refer to previous work that studied the role of the social context, in particular the heterogeneity of phenotypes, on the division of labor in social insects. In particular, it would have been relevant to cite and discuss some of Fewell's studies on the division of labor in associations of ant foundresses or bees. Duarte's theoretical work also contains elements relevant to those found in the present study.

I also regret that the authors did not discuss their results more comprehensively. I would have appreciated more in-depth comments on the mechanisms underlying the observed patterns. In particular, the authors only briefly describe the results of the exploration of their model but without providing any substantial interpretation. As things stand, the argument of the recapitulation found in the mixed groups (Fig. 3) was not enough to convince me that differences in task performance efficiency or changes in demand (lines 182-183) were at work in ant colonies. This certainly opens up some interesting avenues for future research but I believe that additional data should be collected (or presented) to give more credit to this hypothesis and thus grant publication in PloS Biology.

Overall, although I appreciate that the division of labor is a common property of sociality, I doubt that this work will be of interest for a large audience. I would recommend the authors to submit this work to a more specialized journal.

As minor comments, the authors should mention what were the initial conditions of each simulation in terms of stimulus levels. Also, it would also have been useful to provide information on the evolution of the stimulus level for each task when manipulating demand (Fig 3), as the authors wrote "when the demand was so high, the least effective type could not keep up with the demand on its own" (lines 188-189). With a high level of demand for one task, were simulated workers able to cope with the two tasks?

Reviewer #4:

The authors use a clonal ant to test long-standing ideas about the effect of group composition on division labor. They found that the simple, often touted models based only on behavioral thresholds are insufficient to capture the variability of this the observed behavior. When they then added variation in task efficiency and task demand between types they could recapitulate the observed behaviors. 

The story is well-conceived, the research elegantly done and the paper well-written.

How group composition impacts division of labor and group-level efficiency are important topics not just to behavioral biologists, but also to social scientists and employers more broadly. The use of this species to answer this question is novel and solves problems in experimental design that could not be addressed in other ways. The incorporated concepts of task efficiency and stimulus intensity are useful and I hope this model will be applied in other situations. 

Major (not necessarily major, but should be addressed):

*Why is the morphology experiment only done with 8 individuals and 16 times (while the other experiments had 16 individuals, 8 times)? 

*I would have liked more of the model (including equations) in the results section. Understanding how it is built is necessary to appreciate and understand your findings! I would also like more discussion on the implications of the model and how these contagions/amplification relate to existing literature.

Minor:

*I felt that there needed to be a bit more of an opening to the results section. For example, in the first sentence of the result, the authors refer to "the model" but it is not clear which model is being referred to. 

*The phrase "between-type differences" is somewhat confusing. In places it felt like a specific term you were using that needed definition, when eventually I realized it is just a way to express consistent variation between types for a given parameter. It is also only used in the main text, but not in figures and supplement. In many cases one could cut this phrase and the sentences are easier to understand. 

*I would like more discussion. How does this model relate to the many other models that have been developed for division of labor? What can be deduced about forms of behavioral variation and what can be predicted? The authors are frank in that their results vary in all possible directions. What can we take away from this? Would results be different had they analyzed still other genotypes or ages? 

*Age. In the lifespan of these ants, where are the ages used here (are these very old ants, very young, young-middle and old-middle, etc)? 

*The authors should explain more clearly why this work could not be done with "normal" social insects. 

*Can you look at your behavioral data included here and determine task efficiency or behavioral threshold per individual? (more of a curiosity than a request for a change in the manuscript)

Resource Availability: 

*The agent-based models and stats are done in R but so far as I could tell no code was available.

COMMENTS FROM THE ACADEMIC EDITOR:

I'd like the following comments to be addressed:

Efficiency: I could not find a clear description on how efficiency is defined or measured, and this seems really crucial. Moreover, the authors do not consider experience as a determining factor in efficiency, which seems a significant oversight, especially since experience has been shown to generate division of labour in the same species by other authors: https://www.sciencedirect.com/science/article/pii/S0960982207016168

Contagion: I was not sure I understood their notion of behavioural contagion – this should benefit from a clearer definition.

Scholarship: In addition to the above study on clonal raider ants, I agree with one of the referees that Jennifer Fewell's work on threshold models must be cited. The earliest version of the sensory threshold idea and division of labour dates to Francois Huber (1814): Nouvelles Observations sur les Abeilles (Second Edition). There is an English translation by Dadant available on the internet somewhere.

---

## [Decision Letter · Decision Letter 2]

28 Apr 2021

Dear Daniel,

Thank you for submitting your revised Research Article entitled "Response thresholds alone are insufficient to explain empirical patterns of division of labor in social insects" for publication in PLOS Biology. I have now obtained advice from three of the original reviewers and have discussed their comments with the Academic Editor. 

Based on the reviews, we will probably accept this manuscript for publication, provided you satisfactorily address the remaining points raised by the Academic Editor and my data and other policy-related requests.

IMPORTANT: Please address the following:

a) We're aware that the title has been honed in response to comments from the reviewers, but we were wondering whether you could phrase it a bit more straightforwardly, maybe "Response thresholds alone cannot explain empirical patterns of division of labor in social insects"?

b) Please supply a blurb according to the instructions in the submission form.

c) Please address the minor request from the Academic Editor (see the foot of this email).

d) Please address my Data Policy requests further down. Specifically, we need you to supply the underlying numerical values for Figs 1ABC, 2BCDE, 3ABCDE, S1, S2ABCD, S3ABCD, S4ABCDEF, S5ABCD, S6ABCD, S7ABCD, and cite the location of the data clearly in the relevant Fig legends.

We expect to receive your revised manuscript within two weeks. 

*Published Peer Review History*

*Early Version*

Sincerely,

Roli

Senior Editor,

rroberts@plos.org,

PLOS Biology

DATA POLICY:

Regardless of the method selected, please ensure that you provide the individual numerical values that underlie the summary data displayed in the following figure panels as they are essential for readers to assess your analysis and to reproduce it: Figs 1ABC, 2BCDE, 3ABCDE, S1, S2ABCD, S3ABCD, S4ABCDEF, S5ABCD, S6ABCD, S7ABCD. NOTE: the numerical data provided should include all replicates AND the way in which the plotted mean and errors were derived (it should not present only the mean/average values).

DATA NOT SHOWN?

REVIEWERS' COMMENTS:

Reviewer #1:

The revision has dealt adequately with all my concerns.

Reviewer #2:

The authors have revised the manuscript considerably in response to my original comments, which mainly related to points of clarity. All of the points that I raised have been satisfactorily addressed. I congratulate the authors on an interesting and well-formulated study and have no further issues to raise.

Reviewer #4:

Well done. The authors have done a great job incorporating my feedback and the feedback of the other reviewers and editor (excellent points!). It has made the paper more readable and impactful. I especially like the title change and mild shifts in framing. 

I recommend the paper for publication.

COMMENT FROM THE ACADEMIC EDITOR (lightly edited):

I am happy with the revisions the authors have made in response to the referees. I am still not fully convinced by their definition or usage of the term "behavioural contagion" - I'd like to see a bit more on how the authors envisage the spread of information between individuals.

Do the authors have some sort of social learning in mind (cf Alem et al (2016) Associative Mechanisms Allow for Social Learning and Cultural Transmission of String Pulling in an Insect. PLoS Biology 14(10): e1002564. doi:10.1371/journal. pbio.1002564)...

...or the kind of information transfer that happens in animal swarms (Revealing the hidden networks of interaction in mobile animal groups allows prediction of complex behavioral contagion. By: Rosenthal, Sara Brin; Twomey, Colin R.; Hartnett, Andrew T.; et al. PNAS Volume: ‏ 112 Issue: ‏ 15 Pages: ‏ 4690-4695 Published: ‏ APR 14 2015)?

In brief, what are some plausible mechanisms of information spread? This will just take 15 minutes to address, but I'd like to see this discussed.

---

## [Editor Report · Decision Letter 3]

7 May 2021

Dear Daniel,

On behalf of my colleagues and the Academic Editor, Lars Chittka, I'm pleased to say that we can in principle offer to publish your Research Article "Response thresholds alone cannot explain empirical patterns of division of labor in social insects" in PLOS Biology, provided you address any remaining formatting and reporting issues. These will be detailed in an email that will follow this letter and that you will usually receive within 2-3 business days, during which time no action is required from you. Please note that we will not be able to formally accept your manuscript and schedule it for publication until you have made the required changes.

PRESS: We frequently collaborate with press offices. If your institution or institutions have a press office, please notify them about your upcoming paper at this point, to enable them to help maximise its impact. If the press office is planning to promote your findings, we would be grateful if they could coordinate with biologypress@plos.org. If you have not yet opted out of the early version process, we ask that you notify us immediately of any press plans so that we may do so on your behalf.

Thank you again for supporting Open Access publishing. We look forward to publishing your paper in PLOS Biology. 

Sincerely,

Roli 

Roland G Roberts, PhD 

Senior Editor 

PLOS Biology